# Elucidation of divergent desaturation pathways in the formation of vinyl isonitrile and isocyanoacrylate

Wantae Kim[1,5], Tzu-Yu Chen [2,5], Lide Cha[2], Grace Zhou[3], Kristi Xing[3], Nicholas Koenig Canty [2], Yan Zhang [3,4] ✉ & Wei-chen Chang [2] ✉

Two different types of desaturations are employed by iron- and 2-oxoglutarate-dependent (Fe/2OG) enzymes to construct vinyl isonitrile and isocyanoacrylate moieties found in isonitrile-containing natural products. A substrate-bound protein structure reveals a plausible strategy to affect desaturation and hints at substrate promiscuity of these enzymes. Analogs are synthesized and used as mechanistic probes to validate structural observations. Instead of proceeding through hydroxylated intermediate as previously proposed, a plausible carbocation species is utilized to trigger C=C bond installation. These Fe/2OG enzymes can also accommodate analogs with opposite chirality and different functional groups including isonitrile-(D)-tyrosine, N-formyl tyrosine, and phloretic acid, while maintaining the reaction selectivity.

Isonitrile-containing natural products have been identified from terrestrial and marine microorganisms[1-11]. Among these molecules, several of them are used to construct the core of xanthocillin, hapalindoles, paerucumarin, rhabduscin, and byelyankacin (Fig. 1A)[8,10-16]. Biosynthetic gene cluster analysis reveals that a conserved strategy is employed in assemblies of vinyl isonitrile and isocyanoacrylate, e.g. **1**, **2**, and **3**, wherein the first reaction includes a condensation reaction to furnish the isonitrile group[17-19]. In the second reaction, iron- and 2-oxoglutarate-dependent (Fe/2OG) oxygenases catalyze two different types of desaturation reactions to install olefin moiety[17-19]. In **1** and **3**, the reaction includes cleavage of a C-H bond and a decarboxylation, while production of **2** is a formal dehydrogenation.

Fe/2OG oxygenases catalyze a broad array of reactions. Examples include hydroxylation, epoxidation, endoperoxide formation, halogenation, rearrangement and many more[20-25]. In all characterized Fe/2OG enzymes, following an Fe(IV)-oxo species triggered hydrogen atom transfer (HAT) or oxygen atom transfer (OAT), diverse pathways are deployed to affect reaction outcomes. While tremendous efforts have been devoted to engineering Fe/2OG hydroxylases and halogenases to prepare valuable molecules[26-30], using Fe/2OG enzymes for

the production of vinyl isonitrile and isocyanoacrylate moieties remains understudied. In the canonical hydroxylation, the radical generated via the HAT step is captured by the Fe(III)-OH, resulting in C−O bond formation and regeneration of Fe(II)[31,32]. This step is also known as oxygen-rebound (Fig. 2)[33]. While employing the canonical oxygen-rebound intermediate may serve as a common pathway where dehydrogenation or decarboxylation is exercised to eliminate the preinstalled hydroxyl group[34], another pathway involving a cation is conceivable wherein formation of the benzylic cation triggers $CO_2$ or proton removal, respectively (Fig. 2).

Herein, we purify and reconstitute activity of two Fe/2OG enzymes, PvcB and PIsnB, that catalyze **2** and **3** formation in the biosynthesis of paerucumarin and rhabduscin to orchestrate concise chemistries and to establish governing factors utilized for vinyl isonitrile and isocyanoacrylate moieties installation. Additionally, we obtain a high-resolution structure of the substrate-bound PIsnB, wherein substrate flexibility is proposed and investigated using analogs. While the active site and substrate binding of PIsnB and PvcB are highly similar, different reaction products (**2** and **3**) are generated by PvcB and PIsnB, respectively. To explore the molecular

[1]McKetta Department of Chemical Engineering, University of Texas, Austin, TX, USA. [2]Department of Chemistry, NC State University, Raleigh, NC, USA. [3]Department of Molecular Biosciences, University of Texas, Austin, TX, USA. [4]Institute for Cellular and Molecular Biology, University of Texas, Austin, TX, USA. [5]These authors contributed equally: Wantae Kim, Tzu-Yu Chen. ✉e-mail: jzhang@cm.utexas.edu; wchang6@ncsu.edu

**Fig. 1 | Fe/2OG enzymes catalyze desaturation reactions deployed in the natural products and mechanistic probes used herein. A** Hapalindole U, paerucumarin, rhabduscin, and byelyankacin are biosynthesized via **1**, **2**, or **3** as key intermediates. Iron- and 2-oxoglutarate-dependent enzymes catalyze decarboxylation-assisted desaturation and formal dehydrogenation to install vinyl isonitrile (**1** and **3**) and isocyanoacrylate (**2**) groups, respectively. The position of desaturation is highlighted in red. **B** Substrate analogs (**5–8**) and mechanistic probes (**9–10**) used in this study.

determinants and to elucidate the plausible pathway accounting for the observed reaction selectivity, substrate analogs (**5-8**) and mechanistic probes (**9-10**) are prepared and investigated in vitro. Instead of employing the canonical oxygen-rebound pathway, a benzylic carbocation may potentially be equipped for use en route to **2** and **3** formation. Furthermore, sequence alignment analysis reveals that vinyl isonitrile and isocyanoacrylate forming enzymes contain divergent sequences at conserved positions, hinting at reaction selectivity. An orthologue, PIsnB-Ah, with an uncharacterized function is expressed and tested to validate this hypothesis.

## Results and discussion

### In vitro characterization of PIsnB- and PvcB-catalyzed reactions

We carried out in vitro assays to establish the reaction profile of the PvcB- and PIsnB-catalyzed reactions. Both enzymes were expressed as $N$-His$_6$-tagged proteins. The reactions include reconstituted enzyme/ substrate **4**/2OG/ascorbate in a ratio of 1/5/15/10, with a final concentration of 0.1 mM of reconstituted enzyme in Tris-HCl (50 mM, pH 7.7). The enzymatic reactions were analyzed using liquid chromatography coupled mass spectrometry (LC-MS). In the PIsnB-catalyzed reaction, the major product with an $m/z$ value corresponding to the vinyl isonitrile was detected ($m/z$ 190.1→144.1, Fig. 3). Identity of this peak was verified using the synthetic standard of **3** (Fig. 3). On the other hand, formation of a peak with an $m/z$ value matching isocyanoacrylate (**2**) was observed in the PvcB-catalyzed reaction ($m/z$ 190.1→188.1, Fig. 3). In both cases, a minor peak with an $m/z$ value consistent with hydroxylation was also detected ($m/z$ 190.1→206.1).

Notably, **2** is not very stable and readily decomposes to **11** through hydration of the isonitrile group (Supplementary Fig. 1). This observation is in accordance with the literature where **2** could only be characterized by derivatization[34].

### X-ray crystal structure of substrate-bound PIsnB

To understand the molecular insight that leads to the observed selectivity of PvcB and PIsnB, we conducted a structural investigation of PIsnB with its substrate (**4**). Since the oxidation state of iron can be altered under aerobic conditions, which may lead to heterogeneity affecting crystal growth, PIsnB was reconstituted with various metal ions to obtain stable complexes. Out of a dozen metal ions screened, manganese (Mn) improves the melting temperature ($T_m$) of PIsnB by more than ten degrees, suggesting that Mn ion effectively stabilizes the protein, possibly through coordination to the iron-binding site (Supplementary Fig. 2A). The inclusion of compound **4** with PIsnB also has a stabilizing effect by improving the $T_m$ (35→40 °C) (Supplementary Fig. 2A). The ternary complex (PIsnB•Mn•**4**) was prepared for structure determination by co-crystallization approach. Specifically, the substrate-bound PIsnB was assembled by incubating recombinant protein at ~1 mg/mL with 2.5 mM of Mn ion and 2 mM of **4** for 3 h. The mixture was then concentrated to ~10 mg/mL of protein and subjected to crystallization screening. The diffraction data were collected to a resolution of 1.98 Å, with each asymmetric unit containing one molecule of PIsnB in the space group P2$_1$2$_1$2$_1$. The final model shows that PIsnB exists as a monomer which is consistent with the oligomeric state determined by gel filtration (Supplementary Fig. 3). Analogous to

**Fig. 2 | Plausible reaction mechanisms account for Fe/2OG enzyme-catalyzed hydroxylation and desaturation. A** Canonical hydroxylation is initiated by an Fe(IV)-oxo species triggered hydrogen atom transfer (HAT). It is then followed by oxygen-rebound to complete hydroxylation. The HAT site and Fe(IV)-oxo are highlighted in blue and red, respectively. **B** In the Fe/2OG enzymes, i.e., PlsnB and other Fe/2OG enzymes, PlsnB has two four-stranded β-sheet in the center, a.k.a., jelly roll or Swiss roll fold (Supplementary Fig. 4A)[35]. The metal ion is coordinated by His101, Asp103, His250, and three water molecules (Fig. 4A).

The omit map calculated using protein sequence alone reveals a significant positive density close to the metal center (Supplementary Fig. 4B). We first considered molecules used during *Escherichia coli* growth and crystallization and modeled 2-(N-morpholino)ethane-sulfonic acid and L-tyrosine, but significant negative and positive density indicates that they cannot account for the observed density (Supplementary Fig. 4C, E). On the other hand, the chemical structure of **4** is consistent with the shape of the density (Fig. 4B and Supplementary Fig. 4D). Iterative refinement of **4** shows a fairly good fit of the ligand although the density of one of the *m*-carbons of the benzene ring is slightly weaker compared to the rest of the ligand (average B factor of the phenyl ring is 43.1 Å$^2$ compared to an average of ~38.6 Å$^2$ for the rest of the ligand) (Supplementary Fig. 4). The benzene ring of **4** locates in a hydrophobic pocket formed by Tyr106, Ile159, Trp102, and Met105 (Fig. 4C). While the ample space of the hydrophobic pocket allows the rotation of the benzene ring, the π-π stacking of the phenyl ring of **4** and His101 may result in a preferred orientation as the observed in the crystal structure. Furthermore, the hydrophobicity of the pocket recognizing the phenyl ring of **4** has an effect for substrate recognition. When we alter the hydrophobicity of the pocket by mutating Met105-Tyr106-Lys107 to Ala-Phe-Ala, the substantial improvement of T$_m$ upon ligand incorporation observed in wild-type protein is no longer retained (Supplementary Fig. 2B), suggesting the loss of ligand binding (Supplementary Fig. 2B). Notably, no obvious interactions for the *para*-hydroxyl group can be identified, thus suggesting other functional groups, e.g., fluoride (**7**), can be accommodated in the active site pocket.

**Potentially catalytic role of the capping loop in PlsnB**
The carboxylate moiety of **4** is well-positioned by hydrophilic interactions. Namely, the carboxylate is anchored by Arg265, which has a

PvcB, catalyzed desaturation reactions using **4** as the substrate, following HAT step, a hydroxylated intermediate or a benzylic cation may serve as a common species to afford **2** and **3** via deprotonation (pathway a, blue arrow) and decarboxylation (pathway b, red arrow), respectively. Other alternative mechanisms are summarized in the supplementary information (Supplementary Fig. 23).

cation-π interaction with Trp74 with a distance of 3.5 Å to the indole ring (Fig. 4D). It also forms hydrogen bonds with Gly104 and an iron-coordinated water molecule (Fig. 4D). It is worth mentioning that a loop region (I159-V162) is disordered in the ternary complex structure. Sequence alignment analysis suggests that this region is not conserved among Fe/2OG enzymes that catalyze vinyl isonitrile or iso-cyanoacrylate formation (Supplementary Fig. 5). To evaluate the potential role of this capping loop, we mutated this loop to the corresponding loop in PvcB (I159R_N160S_K161A) and characterized the product profile using LC-MS. Under current experimental conditions, while this variant is substantially less active (~20% of **3** was produced compared to the wild-type enzyme), the product distribution is not altered (Supplementary Fig. 6). Therefore, the capping loop likely provides a hydrophobic environment to maintain effective catalysis but does not affect the selectivity.

**The active site reveals a plausible strategy to prevent isonitrile chelation to the iron center and hints at the substrate flexibility**
While isonitrile is a potent and a common ligand to iron[36–39], coordination of isonitrile to the iron in the active site inactivates PlsnB as demonstrated by Mössbauer spectroscopy previously[40]. The substrate-bound structure reveals how PlsnB prevents isonitrile chelation to the iron while maintaining catalytic efficiency. Since the carboxylate group has hydrogen bonds with Gly104 and an iron-coordinated water, such orientation enforces the isonitrile moiety to point away from the iron center (Fig. 4D). The position of the isonitrile group is further illustrated when we placed L-tyrosine in the active site density (Supplementary Fig. 4E). A positive density suggests that the additional density correlated to the methyl group of NC moiety. Unlike the strongly anchored carboxylate group, the isonitrile moiety of the substrate is much less restricted in space (Fig. 5A). No apparent interaction can be identified within 5 Å of the isonitrile group. The ample space accommodating the isonitrile group and the well-positioned carboxylate group in the active site suggest that other groups, e.g., formyl group (**6**), proton (phloretic acid, **8**) and the

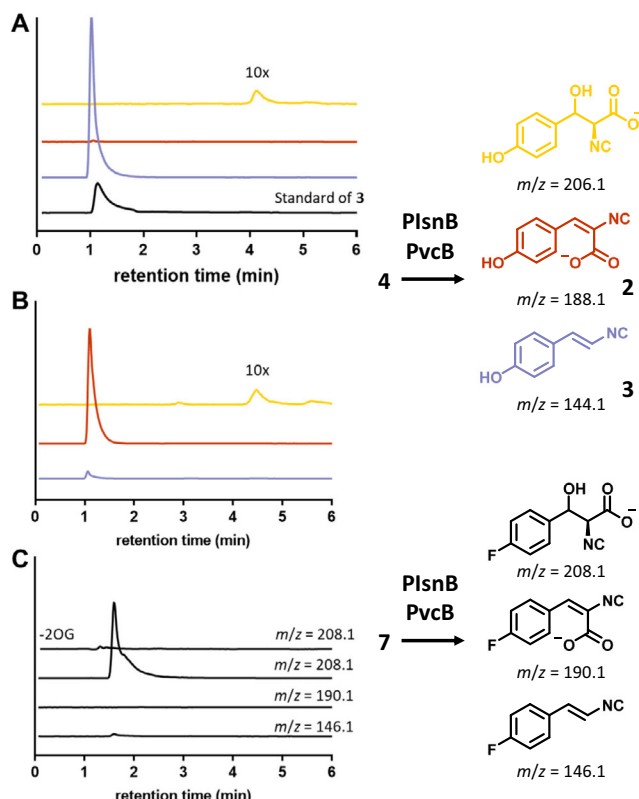

**Fig. 3 | PIsnB and PvcB catalyze chemically divergent desaturations.** LC-MS chromatograms of PIsnB- and PvcB-catalyzed reactions. **A** In the PIsnB reaction, conversion of **4** to vinyl isonitrile **3** was identified (*m/z* 190.1→144.1, purple). **B** Isocyanoacrylate **2** was detected as the dominant product in the PvcB-catalyzed reaction (*m/z* 190.1→188.1, red). In both PIsnB- or PvcB-catalyzed reactions, a minor hydroxylation product (*m/z* 206.1, yellow) was detected. The intensity of the hydroxylated peak is magnified 10-fold for clear visualization. **C** When the *para*-substituent is replaced by fluoride (**7**), PvcB and PIsnB only catalyze hydroxylation (*m/z* 192.1→208.1). Products associated with desaturations (*m/z* 192.1→146.1 or 190.1) were not detected.

substrate analog with the opposite chirality (**5**), may also fit into the active site. Indeed, the *N*-formyl moiety has been identified in several natural products, including fumiformamide and melanocin E[10,11]. To validate this observation, we modeled **6** in the active site of PIsnB. The *N*-formyl group does not give rise to an obvious steric clash (Fig. 5B). Furthermore, the molecular modeling using **5**, **7** and **8** indicates that all hydrophobic interactions at the phenyl ring and the hydrophilic interaction at the carboxylate group are well-preserved (Supplementary Fig. 7). These observations suggest that **5**-**8** could serve as potential substrates for PIsnB.

## LC-MS and ¹³C-NMR reveal substrate flexibility of PIsnB and PvcB

Analogs **5** and **6** were chemically prepared and incubated with PIsnB and PvcB to validate the observations revealed by in silico analysis. Indeed, **5**, the analog with the opposite chirality of **4**, was converted to **2** and **3** by PvcB and PIsnB, respectively (m/z: 190.1→188.1 (**2**) and 144.1 (**3**). Supplementary Fig. 8). Furthermore, **11** and **12** were produced at the expense of **6** by PvcB and PIsnB (m/z: 208.1→206.1 (**11**) and 162.1 (**12**), Supplementary Figs. 8B and 9). Additionally, PIsnB can also catalyze 4-vinylphenol formation using phloretic acid **8** as the substrate (Supplementary Fig. 10). Thus, LC-MS results support the aforementioned observation and demonstrate the substrate flexibility and the reaction selectivity of PIsnB and PvcB. As shown in Supplementary Fig. 8A, similar level of substrate consumption as of the native substrate (**4**) was observed. Under the current reaction conditions (0.1 mM

reconstituted enzyme), **4**, **5**, and **6** were consumed ~0.96, 0.80, and 0.45 mM in the PIsnB-catalyzed reactions. In PvcB, ~0.82, 1.0, and 0.74 mM of **4**, **5**, and **6** consumption were detected. To further support the LC-MS results, structures of **11** and **12** are confirmed by ¹³C-NMR using [2-¹³C]-**6** as the substrate. In the PIsnB-catalyzed reaction, three peaks with chemical shifts of 119, 120, and 124 ppm were observed (Fig. 6). Similar to [2–¹³C]-**6** that exhibits two peaks ($\delta = 56.8$ and 61.2 ppm) representing two tautomers in the solution state, **12** shows two peaks at 120 and 124 ppm[41]. The third peak ($\delta = 119$ ppm) most likely originates from the other stereoisomer with the *cis*-geometry of the double bond. On the other hand, in the PvcB-catalyzed reaction, two peaks at 132 and 133 ppm are consistent with formation of **11** (Fig. 6). A minor peak at 120 ppm was also detected. While the configuration, i.e., *cis*- vs. *trans*-isomer, of the newly installed olefin of **11** and **12** remains to be determined, LC-MS and NMR results demonstrated that PIsnB and PvcB can accommodate substrate analogs and affect chemically divergent desaturations. Notably, formation of **3** using **4** and **5** implies that reaction pathway involves the second hydrogen atom abstraction (Supplementary Fig. 23) is less likely wherein changing the chirality from L to D, i.e., **4** to **5**, likely inverts the Cα-H position in the active site (Supplementary Fig. 7).

## Fluorinated substrate analog and hydroxylated probes reveal plausible pathway of the PIsnB- and PvcB-catalyzed desaturation

In the PIsnB-catalyzed reaction, alternation of the *para*-substituent from an electron-donating group, e.g., hydroxyl group, to an electron-withdrawing group, e.g., fluoride, changes the reactivity from decarboxylation-assisted desaturation to hydroxylation, thus weighing against the pathway includes hydroxylation as an intermediate[40]. If a carbocation species is deployed by PvcB, due to electron-withdrawing property of fluoride that destabilizes the carbocation, one would expect **7** to impede isocyanoacrylate formation, thus directing the reaction outcome. In contrast, if desaturation undergoes hydroxylation followed by dehydration as proposed previously or other pathways involving an electron-transfer promoted C–C bond cleavage (Fig. 2 and Supplementary Fig. 23), **7** could decrease product formation, but should not influence product distribution. To test this hypothesis, substrate analog with a fluoride appended at the *para*-position (**7**) was synthesized. The enzymatic reactions using **7** was carried out. As shown in Fig. 3C and Supplementary Fig. 8A, under the similar conditions, while similar level of the substrate consumption was detected (~0.96/0.60 and 0.82/0.50 mM substrate (4/7) consumption in PIsnB and PvcB, Supplementary Fig. 8A), only the peak with an *m/z* value corresponds to hydroxylation was detected in both PIsnB and PvcB (Fig. 3C). Furthermore, we compared the active sites of PIsnB and PvcB[16,34] by superimposing the two protein structures and modeled **4** into the PvcB structure (Fig. 5C, D). Residues interacting with the substrate (**4**) and the iron center are highly conserved in PIsnB and PvcB (Figs. 4D and 5D). All interactions required for ligand binding are maintained, including the histidine imidazole sidechain stacking with the substrate's aromatic ring and the hydrophilic interactions anchoring the carboxylate group (Fig. 5D). In addition, modeling of **7** in the active site of PIsnB reveals that **7** and **4** bind analogously (Supplementary Fig. 7). While formation of a hydroxylated product using **7** is consistent with the mechanism involves the intermediacy of cation, it could also be caused by dissociation of that species from the active site prior to dehydration or the fluoride-substitution may induces the departure of the hydroxylated compound (**10**). To further elucidate the reaction pathway, analogs (**9** and **10** with the *para*-substituent being H or F) were prepared and investigated. Under the condition of enzyme to substrate ratio of 1:20 with the final enzyme concentration of 100 μM, no obvious new peak can be detected (Supplementary Fig. 13A). Another possibility is that **9** and **10** cannot enter the active site. We carried out the competition experiment by incubating the

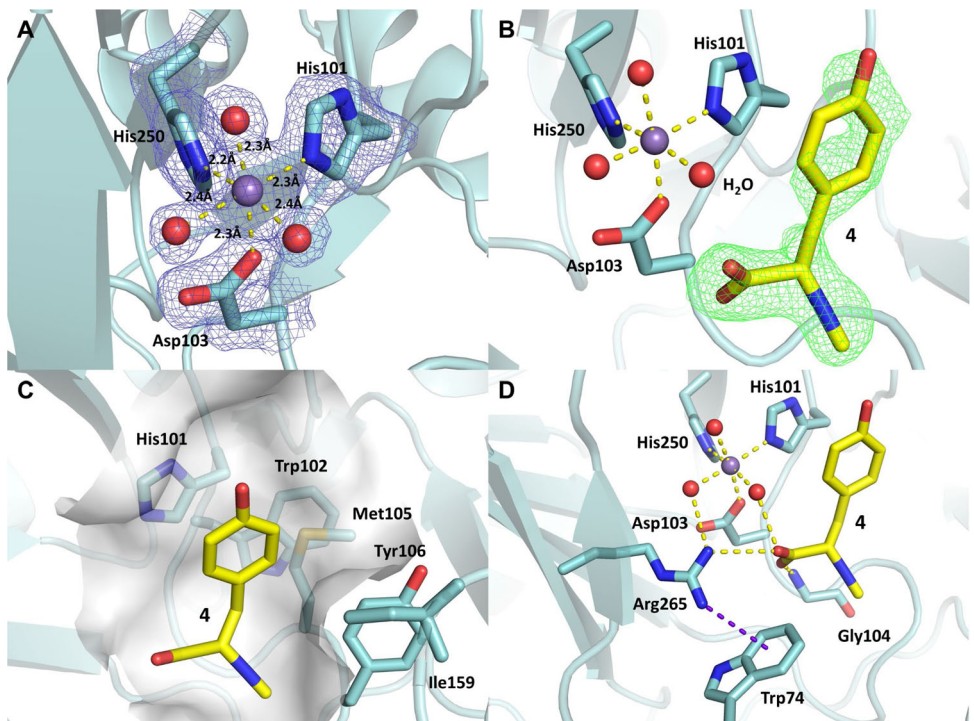

**Fig. 4 | Structure of PlsnB•Mn•4. A** 2FoFc map of the metal center of PlsnB•Mn•4. Dash lines show the coordination of the metal with the distances labeled. The 2FoFc composite map is shown by blue mesh contoured to 1.5σ. **B** FoFc omit map of the PlsnB active site upon incorporating the native substrate (**4**) in co-crystallization experiment, contoured to 3σ and shown in green mesh. The omit map was calculated with no ligand information to avoid phase bias. The chemical structure of the substrate was then superimposed onto the density for consistency comparison. **C** The hydrophobic pocket close to the aromatic side chain of **4** is shown as semitransparent surface. **D** Hydrophilic interactions of the carboxylate group of **4** with Arg265, Gly104, and a water molecule. Arg265 has a cation-π interaction with Trp74.

native substrate (**4**) with **9** or **10** to assess later possibility. In comparison with the reaction without adding **9** or **10** (Supplementary Fig. 8A), the reduced activity with ~1/2 and 1/7 of substrate (**4**) consumption was detected in PlsnB. Analogously, ~1/3 and 1/9 of substrate (**4**) consumption was observed in PvcB, thus suggesting **9** and **10** can enter the enzyme active site of PlsnB and PvcB (Supplementary Fig. 13B). Taken together, while we cannot completely rule out the pathway includes a hydroxylated intermediate, these results support the intermediacy of a benzylic carbocation in PlsnB and PvcB-catalyzed desaturation. While pathways include oxygen-rebound and two sequential hydrogen atom abstraction processes have been included in several Fe/2OG enzymes catalyzed desaturation through in vitro as well as computational studies[34,42–50], our results imply that a benzylic cation is likely utilized to affect decarboxylation and deprotonation in the PlsnB- and PvcB-catalyzed chemically divergent desaturations (Fig. 2).

**Sequence comparison helps forecast the reaction selectivity**

Highly similar active sites but different reactivities deployed by PlsnB and PvcB suggest that other residues not identified by structural comparison might be important for the observed selectivity. We carried out a sequence alignment analysis of the Fe/2OG enzymes that have been reported to catalyze vinyl isonitrile and isocyanoacrylate production (Supplementary Fig. 11). The comparison revealed that these enzymes contain divergent sequences at conserved positions. Herein, we annotate enzymes catalyze vinyl isonitrile formation as PlsnB-type and those trigger isocyanoacrylate formation as PvcB-type for clarification. Specifically, in PlsnB-type enzymes, a positively charged residue including lysine or arginine is conserved at the K107 position of PlsnB. In contrast, PvcB-type enzymes have a leucine at this position. Moreover, PlsnB-type enzymes occupy a conserved residue

including a nitrogen-containing side chain such as asparagine or histidine at the N188 position of PlsnB in the downstream region. In contrast, PvcB-type enzymes have a cysteine at this position. This observation provides a simple method to forecast the reaction selectivity, and can be used for the product prediction. To test this hypothesis, a PlsnB-type orthologue from *Aeromonas hydrophila*, referred to PlsnB-Ah, with an unidentified function was purified and investigated. This enzyme shows a similar activity and selectivity as of PlsnB and can accommodate analogs **5** and **6** (Supplementary Fig. 8, 9, and 12). While this analysis provides a simple method to predict reaction selectivity, single-point mutation, including K113L and N194C of PlsnB-Ah, does not alter the product profile (Supplementary Fig. 6), and indicates other residues are also involved in reaction selectivity. Further studies are currently ongoing.

In this work, reaction pathways leading to vinyl isonitrile and isocyanoacrylate formation are revealed using protein structure, molecular modeling, and in vitro assays with substrate and analogs. Our findings suggest that a carbocation species may be deployed to enable decarboxylation and deprotonation. Furthermore, a sequence alignment of Fe/2OG desaturases reveals that divergent sequences at conserved positions are associated with the reaction selectivity. This observation is validated via reconstitution of an uncharacterized enzyme, PlsnB-Ah, activity in vitro. As suggested by the substrate-bound protein structure and the substrate modeling, the ample space used for isonitrile group can occupy other groups. Indeed, all investigated enzymes, i.e., PvcB, PlsnB, and PlsnB-Ah, can accommodate analogs with the opposite chirality and a formyl group, while maintaining reaction selectivity. Taken together, these findings not only uncover the strategies of Fe/2OG enzymes to enable divergent desaturations, but also delineate the diverse chemistries catalyzed by Fe/2OG enzymes in natural product biosynthesis.

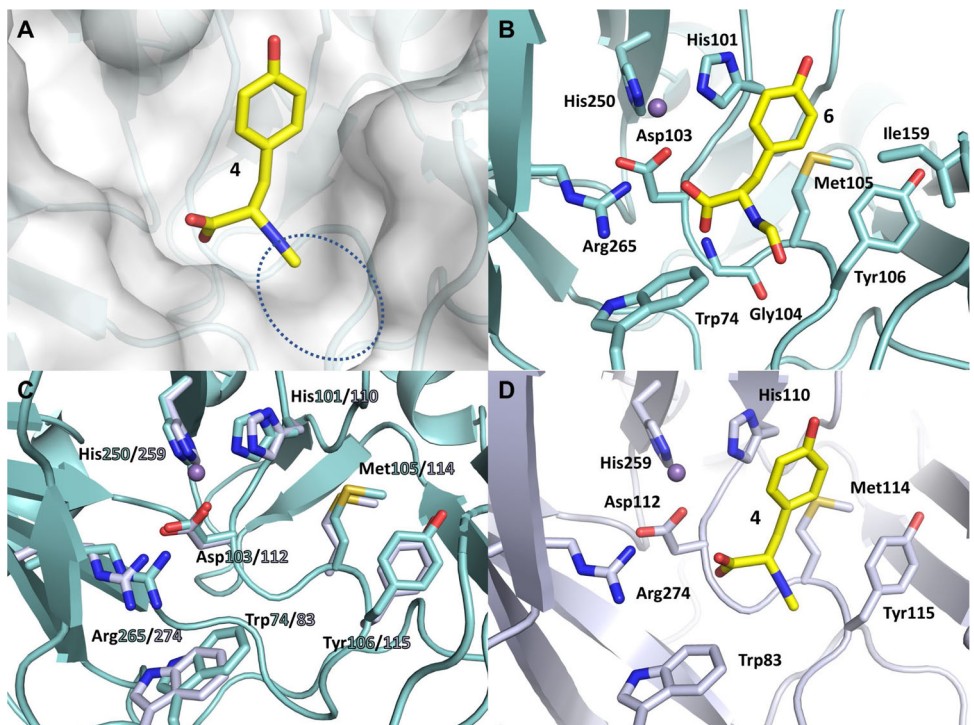

**Fig. 5 | Substrate flexibilities of PlsnB and PvcB. A** Surface representation of **4** in the active site of PlsnB. Compound **4** is shown in the sticks. The circle indicates the region that can accommodate potential functional groups in addition to the iso-nitrile moiety. **B** Modeling of **6** in the active site of PlsnB. Interactions between the phenyl ring, the carboxylate group of **6**, and the residues in the active site are well-preserved. **C** Superimposition of the structures of PlsnB with PvcB (PDB code: 4YLM). Key residues in the active site are shown with the numbering as in PlsnB/PvcB. **D** Modeling of the native substrate **4** in the active site of PvcB, assuming a similar substrate orientation is adopted as of PlsnB•Mn•**4**.

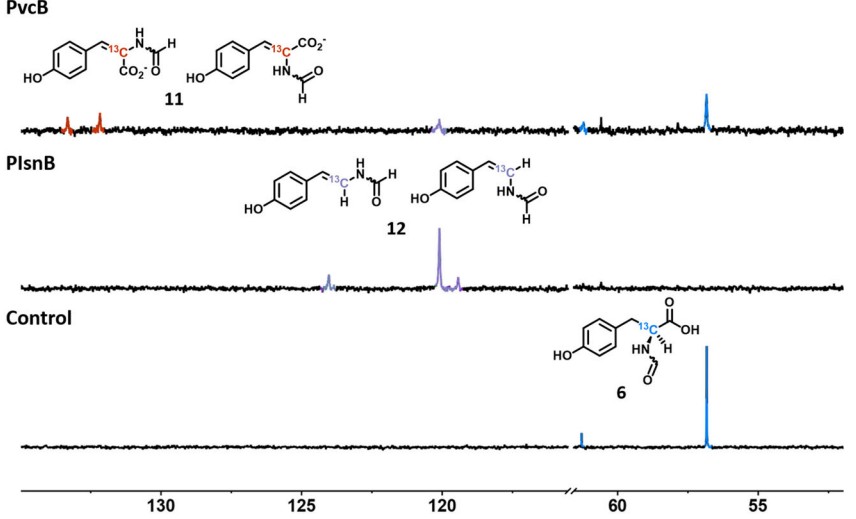

**Fig. 6 | ¹³C-NMR spectra of PlsnB- or PvcB-catalyzed reactions using [2-¹³C]-6.** Compound **12** (purple) with vinyl formyl group was observed in the PlsnB reaction, while *N*-formylacrylate (**11**) (red) was detected as the dominant product in the PvcB reaction. Compounds **6**, **11**, and **12** are highlighted in blue, red, and purple, respectively.

## Methods

### Expression and purification of PlsnB, PvcB, and PlsnB-Ah

The plasmid encoding *pisnB*, *pvcB*, or *pisnB-Ah* gene was transformed into *E. coli* BL21 (DE3) cells (New England Biolabs, MA). A single colony was picked and incubated with 100 mL Luria-Bertani (LB) and 100 μL kanamycin at 37 °C for ~16 h. The cells were used as starting culture for large-scale expression with a volumetric ratio of 1:80 of starting culture to growth media. After inoculation, the cells were growing at 37 °C. Upon optical density at 600 nm (OD₆₀₀) reached of ~0.6, IPTG with final concentration of 1.0 mM was added to the culture. The cells were growing at 18 °C for 15 h before harvesting by centrifugation at 8 °C. To obtain the protein, the cells were suspended in an ice-chilled buffer (100 mM Tris, pH 7.5), and lysed by sonication. The resulting lysate was subjected to centrifugation for 30 min at 22,000 rpm at 4 °C, and the supernatant was loaded onto a Ni-NTA agarose column. The column was washed with six volumes of buffer containing 10 mM imidazole (100 mM Tris, pH 7.5). Subsequently, the desired protein was eluted using buffer containing 250 mM imidazole (100 mM Tris,

pH 7.5). Fractions containing desired protein were observed by sulfate–polyacrylamide gel electrophoresis (SDS–PAGE), and concentrated to a volume of ~3 mL using Pull® 10 K centrifugal filter. The protein solution was then dialyzed against 2 L of buffer with 5 mM EDTA and 100 mM Tris (pH 7.5), and then twice against 2 L of 100 mM Tris buffer (pH 7.5). Protein concentration was determined by UV absorption at 280 nm using a calculated molar absorptivity of 43890, 41160, and 41035 $M^{-1}$ $cm^{-1}$ for PlsnB, PvcB, and PlsnB-Ah, respectively (http://ca.expasy.org). The purities of proteins were shown by SDS–PAGE, and the gel is visualized using Coomassie-stain (Supplementary Fig. 14).

**In vitro assays of PvcB, PlsnB, and PlsnB-Ah catalyzed reactions**
LC-MS was carried out on an Agilent Technologies (Santa Clara, CA) 1200 system coupled to an Agilent Technologies 6120 quadrupole mass spectrometer. The associated Agilent MassHunter and OpenLAB software package were used for data collection and analysis. Detection was performed under electrospray ionization in negative mode (ESI⁻). The drying gas temperature was 350 °C with a nebulizer pressure of 35 psi and flow rate of 12 L/min. The capillary voltage is set to 3000 V. Reaction mixtures were separated on a Merck SeQuant® ZIC®-cHILIC column (150 × 2.1 mm, 3.0 μm particle size). A gradient elution using solvent A (20 mM ammonium acetate aqueous solution) and solvent B (acetonitrile) with a flow rate of 0.35 mL/min was applied. Starting with an isocratic system of 10% solvent A and 90% solvent B, followed by a gradient of 90–60% solvent B from 4 to 8 min. The system was then kept isocratic with 60% solvent B from 8 to 13 min, and then a gradient from 60–90% solvent B was applied from 13 to 17 min. The column was allowed to re-equilibrate for 9 min under initial conditions before subsequent sample injections.

Reactions associated with Fig. 3 are plotted using GraphPad Prism and were performed as described below. Reaction mixtures including enzyme, Fe(II), substrate, 2OG, and ascorbate with the final concentration of 0.12 mM enzyme, 0.1 mM Fe(II), 3 mM 2OG, 0.5 mM substrate, and 2 mM ascorbate with final volume of 200 μL in 50 mM Tris (pH 7.68) were prepared. Reactions were carried out at 4 °C. Reaction samples were quenched using 200 μL of acetonitrile at 10 mins. Prior to LC-MS analysis, all samples were centrifuged at 12,000 g for 30 min to remove the protein.

**Using ¹³C-NMR spectroscopy to follow the enzymatic reactions**
Reaction mixtures containing protein (PlsnB or PvcB), Fe(II), [2-¹³C]-**6**, 2OG, and ascorbate with the final concentration of 0.48 mM protein (PlsnB or PvcB), 0.4 mM Fe(II), 12.0 mM 2OG, 2.0 mM [2-¹³C]-**6**, and 8.0 mM ascorbate with final volume of 600 μL in 50 mM Tris (pH 7.68) were prepared. The reaction mixtures were shaking with a speed of 220 rpm for 17 h at 18 °C. Prior to NMR measurement, 30 μL of DMSO-$d_6$ was added to the reaction followed by centrifugation at 12,000×$g$ for 30 min. The supernatant was then transferred to the NMR tubes. The ¹³C-NMR spectra were recorded using Bruker NEO 700 MHz. The NMR spectra plotted using MestReNova are shown in Fig. 6.

**Crystallization and X-ray structure determination**
In order to identify crystallization conditions for PlsnB, 12 mg/mL of purified PlsnB protein sample was incubated overnight with 2.5 mM $Mn^{2+}$ ion before the screening in sparse matrix with a Phoenix crystallization robotic system (Art Robbins Instruments). After 2-weeks incubation of screening trays at 4 °C, rod shaped crystals appeared in a condition that contains 0.1 M MES (pH 6.5), 0.2 M ammonium sulfate, and 30% PEG 5000MME. This crystallization condition was further optimized by manually setting sitting-drop vapor diffusion experiments with varying pH and precipitant concentration, resulting diffraction-quality crystals. For co-crystallization of **4** with PlsnB protein, 12 mg/mL of PlsnB protein was incubated overnight with 2.5 mM $Mn^{2+}$ ion and 2 mM **4** prior to crystallization setup in an identical condition.

Individual crystals were flash-frozen directly in liquid nitrogen after brief incubation with a reservoir solution supplemented with 30% (v/v) glycerol. X-ray diffraction data were collected at 23-ID-B beamline in Advance Photon Source (Lemont, IL). By using HKL2000[51], X-ray diffraction pattern was processed to 1.98 Å resolution for PlsnB•Mn complexed with **4**. In Phenix software[52], phases were obtained by molecular replacement using a previously obtained PvcB structure as the initial search model (PDB code 4YLM). The molecular replacement solution for PlsnB structure were iteratively built using Coot[53] and Phenix refinement package. The omit map was first calculated, revealing a strong positive density close to the metal binding center. Different chemical compounds used during the expression and purification was manually built inside the density with Coot[53] to calculate the density. The 2FoFc and FoFc maps were with the most likely candidates, as shown in Supplementary Fig. 4C–F. The quality of the finalized crystal structure was evaluated by MolProbity. The final statistics for data collection and structural determination are shown in Supplementary Table 1.

Atomic coordinates and structure factors for the reported crystal structures in this work have been deposited to the Protein Data Bank (PDB) under accession number 7TCL for PlsnB complex with compound **4**.

**Substrate docking**
The binding models for PlsnB with different ligands (**5**–**8**) were obtained by using the complex structure of PlsnB and **4** as an initial reference. The binding mode was then optimized with Maestro (Schrodinger, LLC)[54] which features a minimization routine based on OPLS_2005 Forcefield[55,56]. A substrate binding model of PvcB was conducted in a similar method using apo PvcB structure (PDB code: 4YLM). The initial position of the substrate were obtained by the superimposition of apo PvcB and PlsnB with substrate complex. The model is then subject to energy minimization in Maestro.

## Data availability
Data that support this study are available at the supplementary information. The coordinates are deposited in the Protein Data Bank with PDB accession code 7TCL for PlsnB complex with compound **4**. The DNA sequences encoding PlsnB, PvcB and PlsnB-Ah are from *Photorhabdus luminescens* (WP_011147037.1), *Pseudomonas aeruginosa* PAO1 (AAC21672.1) and *Aeromonas hydrophila* (WP_017765143.1). Data is available from the corresponding authors upon request.

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

## Acknowledgements
This work was supported by the grant from the National Institutes of Health (GM127588 to W.-c.C. and GM104896 and 125882 to Y.Z.) and the Goodnight Early Career Innovator. The authors thank R.Y. Moreno for helpful edits. Crystallographic data collections were conducted at advanced photon sources (BL23-ID-B), Department of Energy (DOE) national user facility.

## Author contributions
W.K., G.Z., and K.X. performed crystallization and X-ray diffraction studies, T.-Y.C., L.C., and N.K.C. performed the biochemical assays and prepared substrate analogs and product standards. Y.Z. and W.-c.C. wrote the manuscript with input from all coauthors.

## Competing interests
The authors declare no competing interests.
