## [Peer Review File · Nature Communications]

Elucidation of divergent desaturation pathways in the formation of vinyl isonitrile and isocynoacrylateREVIEWER COMMENTS

Reviewer #1 (Remarks to the Author):

In this manuscript, the authors explored the two possible desaturation pathways in the formation of vinyl isonitrile and isocynoacrylate by using a series of experiments and simulation. The authors proposed that the Fe/2OG enzymes-catalyzed desaturation undergoes a carbocation species rather than a hydroxylated intermediate. Moreover, the Fe/2OG enzymes can catalyze the substrates with opposite chirality and different functional groups owing to the flexibility of active pocket. These new findings are supported by the experimental evidences, and can provide useful information for understanding the biosynthesis of vinyl isonitrile and isocynoacrylate. In general, the used methodologies are sound, the obtained results are reasonable and noteworthy, thus the present work can provide new insights into the Fe/2OG enzymes-catalyzed desaturation that couples with the decarboxylation, and deserves publication if the authors can address the following concerns of the reviewer.

- 1). Regarding the involvement of carbocation species in the desaturation step: The Fe/2OG oxygenases-catalyzed desaturation is a common reaction, which usually contains two hydrogen abstraction steps, and the first step is triggered by Fe(IV)=O and second step by Fe(IV)-OH. However, if the second step couples with the decarboxylation step, the hydrogen abstraction may be accompanied by an electron transfer from the substrate to the iron center, i.e., the electron transfer promotes the cleavage of C-C bond. Thus, the desaturation may do not undergo a carbocation species. The authors should confirm this point.
- 2). Why the Fe/2OG enzymes show activity toward substrate with the opposite chirality? If the enzymes are active for one type of substrate, another type of substrate is expected to be unreactive, because the hydrogen atom to be extracted in the second step will be in the unfavorable orientation. Please explain it.

Reviewer #2 (Remarks to the Author):

The focus of this manuscript is on the mechanism of dehydrogenation in the formation of vinyl isonitriles and isocynoacrylates by Fe, alpha-ketoglutarate-dependent enzymes. The authors suggest a novel mechanism in which the activated iron-oxo species generates a benzylic cation, which is quenched either by decarboxylation or deprotonation. This is an intriguing proposal, which merits consideration, but the evidence presented by the authors is not very compelling.

The authors base their conclusion on the observation that a p-F-substituted substrate analog for PvcB is converted to the hydroxylated product rather than the dehydrogenated product that is seen with the natural substrate. They argue that this change in product must result from destabilization of a benzylic cation intermediate, therefore, a benzylic cation intermediate must exist.

There are several problems with this interpretation, as well as the data that are presented to support it. In the canonical mechanism, dehydrogenation is proposed to occur via hydroxylated intermediate, which then undergoes dehydration. The authors disfavor a dehydration mechanism because hydroxide is a poor leaving group, but there are myriad enzymes that catalyze exactly that reaction. The fact that analog 7 is converted to a hydroxylated product could very easily be caused by dissociation of that species from the active site, prior to dehydration, i.e., the intermediate normally formed in the reaction dissociates before the dehydration can occur. This interpretation is supported by the inference that 7 is a very poor substrate for PvcB, as shown by the fact that the authors had to incubate the reaction 70 times longer to see product than they did with the other reactions. The authors do not show data to demonstrate how much of the substrate was converted to product in their experiments, and the fact that they conducted their experiments using only a 4-fold excess of substrate over enzyme raises the question of the relevance of their observations. If it takes 10 minutes (or 12 hrs in the case of the PvcB reaction with 7) to generate detectable product, is that reaction really relevant to the natural catalytic mechanism?

The data shown in Figure 2C for the PvcB reaction with 7 does not show a clean peak for product; rather, it appears that another peak with at least equal area is only partially separated from the peak

on which the authors focus.

Mechanistic arguments based on experiments with substrate analogs should include a kinetic characterization of their reactions, so one can evaluate whether the reactions are relevant to the natural, catalytic reaction. In the present case, the extended incubation time required to turn over the substrate analog, as well as the use of near-stoichiometric quantities of substrate and enzyme do not inspire confidence that the reactions observed are useful for determining how the natural reaction occurs.

Reviewer #3 (Remarks to the Author):

This is a nice paper on a novel 2OG-dependent nonheme iron dioxygenase. These are widespread in nature and catalyze a range of chemical reactions. In this case the authors studied a system that is involved in the isonitrile biosynthesis reaction. In order to trap short-lived intermediates they use a substrate-mimic and replace iron by Mn. A few new crystal structures are characterized and product analysis is done for the reactions. Overall this is a nice study that fits the remit of this journal. There have been computational reports on related systems, and I was wondering how the proposed mechanism links to those studies.

Reviewer 1

In this manuscript, the authors explored the two possible desaturation pathways in the formation of vinyl isonitrile and isocynoacrylate by using a series of experiments and simulation. The authors proposed that the Fe/2OG enzymes-catalyzed desaturation undergoes a carbocation species rather than a hydroxylated intermediate. Moreover, the Fe/2OG enzymes can catalyze the substrates with opposite chirality and different functional groups owing to the flexibility of active pocket. These new findings are supported by the experimental evidences, and can provide useful information for understanding the biosynthesis of vinyl isonitrile and isocynoacrylate. In general, the used methodologies are sound, the obtained results are reasonable and noteworthy, thus the present work can provide new insights into the Fe/2OG enzymes-catalyzed desaturation that couples with the decarboxylation, and deserves publication if the authors can address the following concerns of the reviewer.

We thank the reviewer for the positive feedback and our answers to address the reviewer's questions are described below.

1). Regarding the involvement of carbocation species in the desaturation step: The Fe/2OG oxygenases-catalyzed desaturation is a common reaction, which usually contains two hydrogen abstraction steps, and the first step is triggered by Fe(IV)=O and second step by Fe(III)-OH. However, if the second step couples with the decarboxylation step, the hydrogen abstraction may be accompanied by an electron transfer from the substrate to the iron center, i.e., the electron transfer promote the cleavage of C-C bond. Thus, the desaturation may not undergo a carbocation species. The authors should confirm this point.

We appreciate the reviewer for this comment. As the reviewer mentioned, Fe/2OG oxygenase-catalyzed desaturation is a common reaction and has been reported in the several systems. For those involving two C-H bond cleavage, while some Fe/2OG enzymes, e.g., CS2 and VioC involved in clavamate and 3,4-dehydro-L-homoarginine formation, deploy two hydrogen abstraction steps (J. Am. Chem. Soc, 2001, 123, 7388; J. Am. Chem. Soc, 2018, 140 7116), Some other enzymes including NapI and AsqJ used in naphthridinomycin and viridictin formations have been suggested to utilize a possible cation or hydroxylation intermediate (J. Am. Chem. Soc, 2018, 140 7116; Angew. Chem. Int. Ed. 2018, 57, 1831). As the reviewer suggested, if the second step couples with the decarboxylation step, the hydrogen abstraction may be accompanied by an electron transfer from the substrate to the iron center. In this instance, two scenarios can be envisioned: an electron transfer from the benzylic radical or an electron transfer from the oxygen anion of the carboxylate group followed C-C bond cleavage (Scheme S1).

We are inspired by the comment raised by the reviews and by these literature precedent. To provide further mechanistic insight, PlsnB was tested with three analogs (**7**, **9** and **10**, the synthetic schemes and NMR characterization of **9** and **10** are reported in the revised SI). In the instance of a benzylic cation pathway, the electron withdrawing property of fluoride of **7** destabilizes the benzylic cation. To avoid the cation formation, following C-H activation, **7** is expected to redirect the reaction outcome. In contrast, once C-H activation occurs (at the benzylic position), **7** should have lesser influence to affect hydroxylation or the electron transfer promoted C-C bond cleavage. In the PlsnB catalyzed reaction using **7** (Figure 2c), we only detected the hydroxylation product. Additionally, we also tested the possibility of **9** and **10** as the possible reaction intermediates. As shown in Figure S13, no obvious desaturation product can be detected. The corresponding vinyl isonitrile standards (**13** and **14**) were also prepared. Additionally, based on the substrate-bound PlsnB structure, the distance between the carboxylate oxygen to the iron center is ~ 4.8 Å which is probably less suitable for electron transfer. Taken together, these observations are consistent with the intermediacy of a benzylic cation in desaturation.

2). Why the Fe/2OG enzymes show activity toward substrate with the opposite chirality? If the enzymes are active for one type of substrate, another type of substrate is expected to be unreactive, because the hydrogen atom to be extracted in the second step will in the unfavorable orientation. Please explain it.

We appreciate the reviewer for this comment to help us improve the manuscript's clarity. Careful inspection of our crystal structure exhibits extensive interaction between carboxylate of the substrate and protein (Figure 3D). Furthermore, the tyrosyl ring is well-positioned in the hydrophobic pocket in the active site. In D and L substrate-bound models,

these interactions are all maintained when the C β atoms point to different directions and the C α -H atoms point in two directions (as shown below). To provide direct visualization, we added this figure as a new supplementary figure (Figure S7D).

Reviewer 2

The focus of this manuscript is on the mechanism of dehydrogenation in the formation of vinyl isonitriles and isocyanoacrylates by Fe, alpha-ketoglutarate-dependent enzymes. The authors suggest a novel mechanism in which the activated iron-oxo species generates a benzylic cation, which is quenched either by decarboxylation or deprotonation. This is an intriguing proposal, which merits consideration, but the evidence presented by the authors is not very compelling.

The authors base their conclusion on the observation that a p-F-substituted substrate analog for PvcB is converted to the hydroxylated product rather than the dehydrogenated product that is seen with the natural substrate. They argue that this change in product must result from destabilization of a benzylic cation intermediate, therefore, a benzylic cation intermediate must exist.

There are several problems with this interpretation, as well as the data that are presented to support it. In the canonical mechanism, dehydrogenation is proposed to occur via hydroxylated intermediate, which and then undergoes dehydration. The authors disfavor a dehydration mechanism because hydroxide is a poor leaving group, but there are myriad enzymes that catalyze exactly that reaction. The fact that analog **7** is converted to a hydroxylated product could very easily be caused by dissociation of that species from the active site, prior to dehydration, i.e., the intermediate normally formed in the reaction dissociates before the dehydration can occur. This interpretation is supported by the inference that **7** is a very poor substrate for PvcB, as shown by the fact that the authors had to incubate the reaction 70 times longer to see product than they did with the other reactions. The authors do not show data to demonstrate how much of the substrate was converted to product in their experiments, and the fact that they conducted their experiments using only a 4-fold excess of substrate over enzyme raises the question of the relevance of their observations. If it takes 10 minutes (or 12 hrs in the case of the PvcB reaction with **7**) to generate detectable product, is that reaction really relevant to the natural catalytic mechanism?

We appreciate the reviewer for the comments. We agree with the reviewer that in the canonical mechanism deployed by myriad enzymes, dehydrogenation proceeds through

a hydroxylated intermediate, which undergoes dehydration. However, to our knowledge, several literature precedents in Fe/2OG enzymes including AsqJ, CS2, VioC, and NapI suggest that desaturation may undergo alternative pathways (two C-H bond cleavage, as suggested by reviewer #1, or a cation) to enable desaturation (J. Am. Chem. Soc, 2001, 123, 7388; Angew. Chem., Int. Ed. 2014, 53, 12880; Angew. Chem., Int. Ed. 2016, 55, 422; J. Am. Chem. Soc, 2018, 140 7116; Angew. Chem. Int. Ed. 2018, 57, 1831). To clarify ambiguity and provide experimental support, we synthesized two hydroxylation intermediates (**9** and **10**, structures added in Figure 1 and NMR characterization are reported in the Figures S20 and 22) and incubated them with the enzymes. We also carried out enzymatic assays using **7** under the same conditions used for other substrates, as the reviewer suggested. In both PlsnB and PvcB, neither the substrate (**9** and **10**) consumption nor desaturated product formation can be detected (Figure S13) even after prolonged incubation time (16 hours). In excess amount of substrate (enzyme to substrate ratio of 1:20 with the final enzyme concentration of 100 micromolar), we expect to observe desaturated product formation if these analogs serve as a reaction intermediate. We also prepared the standards (**13** and **14**) and accessed the detection limit (~10 micromolar) for the PlsnB reaction. Additionally, as shown in the revised Figure 2C, both PlsnB and PvcB produce the hydroxylated product. In all assays, due to spontaneous hydration of the NC group (Figure S1), we quenched the reaction at a short time point to capture the reaction products. Taken together, these data support the reaction pathway that does not include hydroxylation in the PlsnB and PvcB catalyzed reactions.

The data shown in Figure 2C for the PvcB reaction with **7** does not show a clean peak for product; rather, it appears that another peak with at least equal area is only partially separated from the peak on which the authors focus. Mechanistic arguments based on experiments with substrate analogs should include a kinetic characterization of their reactions, so one can evaluate whether the reactions are relevant to the natural, catalytic reaction. In the present case, the extended incubation time required to turn over the substrate analog, as well as the use of near-stoichiometric quantities of substrate and enzyme do not inspire confidence that the reactions observed are useful for determining how the natural reaction occurs.

We thank the reviewer for this comment to help us increase the clarity of this manuscript. We repeated the experiments wherein we quenched all experiments at 10 minutes, and changed the labeling methods in Figure 2C to clarify the ambiguity. In the revised Figure 2C, compared to the control experiment (without the cosubstrate 2OG), the hydroxylated peak was produced in the presence of 2OG and O₂. As we described above, since there is a spontaneous hydration of NC group which may complicate kinetic characterization, we did not carry out kinetic experiments. Instead, the proposed hydroxylation analogs (**9** and **10**) and the corresponding vinyl isonitrile standards (**13** and **14**) were prepared and investigated with both enzymes. Under current conditions, we did not observe any product formation.

We would also like to thank the reviewer for pointing out the ambiguity that raised from the different reaction conditions. Thus, we have repeated the experiments under same reaction conditions. While we cannot completely rule out the possibility of the

intermediacy of hydroxylation, our results are consistent with the pathway includes a benzylic carbocation. In the revised draft, we added the following sentences "While formation of a hydroxylated product using **7** is consistent with the mechanism involves the intermediacy of a benzylic cation, it could also be caused by dissociation of that species from the active site, prior to dehydration. To further elucidate the reaction pathway, analogs (**9** and **10**) were prepared and investigated. Under the condition of enzyme to substrate ratio of 1:20 with the final enzyme concentration of 100 μ M, no new peak can be detected (Figure S13)"

Reviewer 3

This is a nice paper on a novel 2OG-dependent nonheme iron dioxygenase. These are widespread in nature and catalyze a range of chemical reactions. In this case the authors studied a system that is involved in the isonitrile biosynthesis reaction. In order to trap short-lived intermediates they use a substrate-mimic and replace iron by Mn. A few new crystal structures are characterized and product analysis is done for the reactions. Overall this is a nice study that fits the remit of this journal.

We appreciate the reviewer for the very positive feedback.

There have been computational reports on related systems, and I was wondering how the proposed mechanism links to those studies.

We thank the reviewer for this comment. In several Fe/2OG enzyme catalyzed desaturation reactions, different reaction pathways have been proposed via *in silico* and *in vitro* studies. To address the reviewer's comment and to show readers about current understanding of Fe/2OG enzyme catalyzed desaturation, the following sentence is added in the revised draft "While pathways include oxygen-rebound and two sequential hydrogen atom abstraction processes have been included in several Fe/2OG enzymes catalyzed desaturation through *in vitro* as well as computational studies, our results imply that a benzylic cation is likely utilized to affect decarboxylation and deprotonation in the PIsnB- and PvcB-catalyzed chemically divergent desaturations (Scheme 1)" The references added include J. Comput. Chem. 2006, 27, 740; Nat. Commun. 2018, 9, 1168; J. Biol. Inorg. Chem. 2018, 23, 795; J. Am. Chem. Soc. 2018, 140, 7116; Angew. Chem. Int. Ed. 2018, 57, 1831; J. Am. Chem. Soc. 2019, 141, 20278; Inorg. Chem. 2020, 59, 12218; J. Phys. Chem. A 2021, 125, 1720; Chem. Eur. J. 2022, 28, e202104106.

REVIEWER COMMENTS

Reviewer #1 (Remarks to the Author):

The authors have revised the manuscript according to the comments of the reviewers, now it is suitable for publication in the journal of "Nature Comm."

Reviewer #2 (Remarks to the Author):

This revised manuscript addresses some of the points raised in the first review, but at the end of the day the authors present no definitive evidence in support of their proposed mechanism. Both the intermediacy of a hydroxylated intermediate or the HAT mechanisms suggested by another reviewer are consistent with the results reported by the authors. The authors demonstrate that compound 7 is converted to compound 10 (which, confusingly, is not referred to as such in Figure 2), and then investigate compound 10 as a potential reaction intermediate (line 240) when it has already been established that it is the product of the PvcB and PlsnB reactions. What is the point of incubating synthesized 10 with the enzyme? The failure of compound 9 to react must be interpreted with caution, as the kinetic behavior of exogenously added intermediates is not necessarily straightforward (see *Biochemistry* 1990 29, 3194-3197).

There are still no data presented that indicate how much product is formed with the substrate analogs, and given the apparently extremely slow reaction, it is not clear whether the enzyme even catalyzes multiple turnovers, or if the observed products arise from a single turnover.

If this work is published, the caveats to the authors' interpretations should be presented clearly.

Reviewer #3 (Remarks to the Author):

All issues have been addressed well, publication is recommended.

Reviewer 1

The authors have revised the manuscript according to the comments of the reviewers, now it is suitable for publication in the journal of "Nature Comm."

We thank the reviewer for the positive feedback and support the publication.

Reviewer 2

This revised manuscript addresses some of the points raised in the first review, but at the end of the day the authors present no definitive evidence in support of their proposed mechanism. Both the intermediacy of a hydroxylated intermediate or the HAT mechanisms suggested by another reviewer are consistent with the results reported by the authors. The authors demonstrate that compound **7** is converted to compound **10** (which, confusingly, is not referred to as such in Figure 2), and then investigate compound **10** as a potential reaction intermediate (line 240) when it has already been established that it is the product of the PvcB and PlsnB reactions. What is the point of incubating synthesized **10** with the enzyme? The failure of compound **9** to react must be interpreted with caution, as the kinetic behavior of exogenously added intermediates is not necessarily straightforward (see Biochemistry 1990 29, 3194-3197).

There are still no data presented that indicate how much product is formed with the substrate analogs, and given the apparently extremely slow reaction, it is not clear whether the enzyme even catalyzes multiple turnovers, or if the observed products arise from a single turnover.

If this work is published, the caveats to the authors' interpretations should be presented clearly.

We appreciate the reviewer for the comments and we have carried out the substrate consumption and the competition experiments to support that both enzymes can carry out multiple turnovers using the substrate (**4**) as well as analogs (**5-7**), and the hydroxylated analogs (**9** and **10**) can effectively reduce the native substrate consumption. The figures associated with experiments are included in the revised SI (Fig. S8A and S13B) and are also shown below. Observation of multiple turnovers of substrate analogs and different reaction outcomes (e.g. desaturation vs. hydroxylation) provide experimental support to hint at the plausible reaction mechanism. Compared to **4-6**, only analog **7** results in hydroxylation as the major product. This result suggests that the substituent at the para-position affects the reaction outcome and weighs against the pathway that includes hydroxylation. If a hydroxylated intermediate is included, we expect all analogs should produce the corresponding hydroxylated intermediate. While it is less likely, another possibility is that the fluoride-substitution induces the departure of the hydroxylated compound. To test this possibility, we synthesized **10**. Additionally, we also prepared **9**, wherein a proton is attached to the para-position to assess the possibility that fluoride at the para-position may play a role in inducing the hydroxylated intermediate departure from the enzyme active site. As the reviewer suggested, the failure of using **9** must be interpreted with caution. It is possible that **9** is not the intermediate while produced in the enzymatic reaction. On the other hand, **9**

may not be able to enter the active site of the enzymes. We carried out the completion experiment to test the latter possibility. In this instance, we expect that **9** should not affect the native reaction. In both PlsnB and PvcB, the reduced activity with $\sim 1/2$ and $1/3$ of substrate (**4**) consumption was detected, thus suggesting **9** can enter the enzyme active site. Moreover, consumption of **4** was further reduced to $\sim 1/7$ and $1/9$ in the presence of **10**, which suggests that **10** can also get into the active site. Along with the observations that incubating **9** and **10** with enzymes did not result in desaturation, these experiments support that the PlsnB and PvcB catalyzed reactions are less likely involving hydroxylation as the intermediate. From our point of view, the logical explanation to account for these observations, i.e. fluoride substitution changes the reaction outcome to hydroxylation, is consistent with the carbocation species formation during the desaturation. Because of the electron-withdrawing property of the fluoride that destabilizes the carbocation, the reaction is redirected to hydroxylation.

Figure S8A. Substrate consumption of **4**, **5**, **6** and **7**. From the bottom to top, traces show the substrate peak at the conditions of PlsnB without 2OG (-2OG), PlsnB and PvcB. The substrates have the m/z value of 190.1, 190.1, 208.1 and 192.1 for **4**, **5**, **6** and **7**, respectively.

Figure S13B. Substrate consumption of **4** in the competition experiments using **9** or **10** (**4+9** or **4+10**). The substrate (**4**) has the m/z value of 190.1.

The following changes are added to in the revised draft.

In the section of “LC-MS and ^{13}C -NMR reveal substrate flexibility of PlsnB and PvcB”, “As shown in Fig. S8A, similar level of substrate consumption as of the native substrate (**4**) was observed. Under the current reaction conditions (0.1 mM reconstituted enzyme), **4**, **5**, and **6** were consumed ~ 0.96 , 0.80 , and 0.45 mM in the PlsnB catalyzed reactions. In PvcB, ~ 0.82 , 1.0 , and 0.74 mM of substrates consumption were detected.

In the section of “fluorinated substrate analog and hydroxylated probe reveal plausible reaction pathway”, “While formation of a hydroxylated product using **7** is consistent with the mechanism involves the intermediacy of cation, it could also be caused by dissociation of that species from the active site prior to dehydration or the fluoride-substitution may induces the departure of the hydroxylated compound (**10**).”

In the same section, “Another possibility is that **9** and **10** cannot enter the active site. We carried out the competition experiment by incubating the native substrate (**4**) with **9** or **10** to assess this possibility. In comparison with the reaction without adding **9** or **10** (Fig. S8A), the reduced activity with $\sim 1/2$ and $1/7$ of substrate (**4**) consumption was detected in PlsnB. Analogously, $\sim 1/3$ and $1/9$ of substrate (**4**) consumption was observed in PvcB, thus suggesting **9** and **10** can enter the enzyme active site of PlsnB and PvcB. (Fig. S13B)”

Additionally, we have made the following change in the revised draft in response to the review’s comment about “The authors demonstrate that compound **7** is converted to compound **10** (which, confusingly, is not referred to as such in Figure 2),...”. In the section of “fluorinated substrate analog and hydroxylated probe reveal plausible reaction pathway”, a following sentence is now added.

“To test this hypothesis, substrate analog with a fluoride appended at the *para*-position (**7**) was synthesized. The enzymatic reactions using **7** was carried out. As shown in Fig. 2C and S8A, under the similar conditions, while similar level of the substrate consumption was detected ($\sim 0.96/0.60$ and $0.82/0.50$ mM substrate (**4/7**) consumption in PlsnB and PvcB, Fig. S8A), only the peak with an *m/z* value corresponds to hydroxylation was detected in both PlsnB and PvcB (Fig. 2C).”

Per reviewer’s comment “There are still no data presented that indicate how much product is formed with the substrate analogs, and given the apparently extremely slow reaction, it is not clear whether the enzyme even catalyzes multiple turnovers, or if the observed products arise from a single turnover”, we carried out the substrate consumption experiments. The results reveal that both enzymes can catalyze multiple turnovers of all analogs. As of the product formation in the PlsnB and PvcB catalyzed reactions, the ratio **2**, **3** and hydroxylation in the presence of **4** or **5** as the substrate, and the ratio **11**, **12** and hydroxylation in the in the presence of **6** is shown in the figure S8B. For analog **7**, the only detected product is the hydroxylated compound (Figure 2C). From our point of view, the ratio should reflect how much products are formed associated with the substrate analogs.

Reviewer 3

All issues have been addressed well, publication is recommended.

We appreciate the reviewer for the very positive feedback and recommendation for publication.

REVIEWERS' COMMENTS

Reviewer #2 (Remarks to the Author):

The additional experiments and results reported in the revised manuscript strengthen it, and publication of this version is appropriate.